# Adsorption and Self-Diffusion of R32/R1234yf in MOF-200 Nanoparticles by Molecular Dynamics Simulation

**Biyu Jing [1], Di Xia [2] and Guoqiang Wang [1],***

[1] Key Laboratory of Low-Grade Energy Utilization Technologies and Systems, Ministry of Education, School of Energy and Power Engineering, Chongqing University, Chongqing 400030, China

[2] Power China Sichuan Electric Power Engineering Co., Ltd., Chengdu 610041, China

* Correspondence: ggglxk@cqu.edu.cn; Tel.: +86-(023)65102473

**Abstract:** The thermophysical properties of a refrigerant can be modified via adding metal organic frameworks (MOF) to it. Understanding the adsorption–diffusion process of the mixture in MOFs at the molecular level is important to further improve the efficiency of the organic Rankine cycle. The adsorption and diffusion of R32/R1234yf in MOF-200 was investigated by molecular dynamics simulation in the present work. The results show that the number of adsorbed molecules of R32 in MOF-200 per unit mass is higher than that of R1234yf in the pure fluid adsorption system. The adsorption capacity of the mixture is lower than that of a pure working medium due to competitive adsorption. For both pure and mixed refrigerants, the adsorption heat of R32 in MOF-200 is smaller than that of R1234yf. Compared with R1234yf, the self-diffusion coefficient of R32 in MOF-200 is larger because of the lower diffusion activation energy.

**Keywords:** refrigerants; adsorption; diffusion; nanofluids; molecular dynamics



## 1. Introduction

With the shortage of fossil energy sources and the aggravation of environmental pollution [1], the importance of energy development and utilization is increasingly prominent. In fact, there is a large amount of low-grade energy that can be recovered from natural and industrial production, including solar energy, geothermal energy and waste heat [2–6]. The organic Rankine cycle (ORC), which is an effective approach to utilize low-grade energy [7,8], employs organic refrigerants as a working medium of the Rankine cycle. However, one of the drawbacks for ORC is the relatively low efficiency. Therefore, it is significant to resort to various measures to improve the operating efficiency of ORC [9].

It was found that adding a certain amount of nanoparticles into the working medium can enhance the thermophysical properties of the working medium [10–12], thus improving the ORC efficiency. Therefore, McGrail et al. [13] proposed adding porous metal-organic nanoparticles into the working medium, which can improve the thermal properties of the working medium, further enhancing the efficiency of ORC. MOFs are micro/nano-porous materials consisting of inorganic metal centers and organic ligands, which have designable pore size, topological structures and chemical functionality [14–17]. Compared with other traditional porous materials (silicate, carbon and zeolite), the distinctive composition of MOFs makes them have better adsorption performance [18], preferentially serving as carriers of organic working fluids. The adsorption behaviors of MOF adsorbents for ORC have been widely investigated. Hu et al. [19,20] investigated the energy storage properties of $CO_2$, R1234yf, R1234ze(Z), R134a, R32 in MOF-74 and MOF-5. Zheng et al. [21] investigated the adsorption performance of HFC-134a in MOFs, and the results showed that the HFC-134a/MOF combination has a good prospect in adsorption cooling applications. Barpaga et al. [22] used a combination of experiments and simulations to discuss the adsorption of HFC-134a to better understand the use of equilibrium isotherms and enthalpy interactions. The results showed that the simulated isotherms of HFC-134a in Ni-MOF-74 at low pressure

match well with the experimentally measured isotherms, and Cr-MIL-101 has a stronger interaction with HFC-134a during the initial stage of adsorption. García et al. [23] simulated the adsorption performance of refrigerants in 40 experimentally available MOFs, thus providing a guide to determine the best MOFs/working fluids pair for the application in adsorption air-conditioning. According to previous works, MOF-200 has good adsorption performance on fluid due to its large pore size, high BET and Langmuir specific surface area [24,25]. Thus, MOF-200 was selected as the adsorbent in this paper. Previous studies mainly focused on the discussion of the adsorption phenomenon of pure fluids in MOF, and there were few reports on the study of mixtures in MOF.

The dynamic properties of adsorbed molecules are also important. Diffusion properties are often used to characterize the dynamic behavior of fluids in microporous materials. Since it is difficult to accurately measure the diffusion coefficients of guest molecules inside MOFs [26–29], experimental studies of the diffusion of MOFs are limited [30], and therefore, computational simulations play an essential role in discussing the kinetic properties of MOFs. Skoulidas et al. [31] studied the self-diffusion and transport diffusion of several MOFs with pore loading at room temperature using equilibrium molecular dynamics (MD). They determined self-diffusivity, modified diffusivity, and transport diffusivity as a function of the pore load at room temperature. Ford et al. [32] investigated the relationship between self-diffusion and chain length of hydrocarbon in IRMOF-1 by molecular simulations and NMR experiments. The results showed that the self-diffusivity of hydrocarbons in IRMOF-1 is similar to those in the pure liquid phase, but one order of magnitude higher. Farzi et al. [33] performed an MD simulation to study the diffusion behavior of acetylene molecules in MOF-508a and MOF-508b. The results showed that the self-diffusion of acetylene increased with temperature and loading. Compared with MOF-508b, acetylene diffuses more easily in MOF-508a. Liu et al. [34] investigated the adsorption and diffusion of benzene in Mg-MOF-74 by molecular simulation. The results indicated that Mg-MOF-74 is promising for the capture and separation of benzene. Erhan et al. [35] first reported the transport characteristics of $CH_4/H_2/CO_2$ mixture in the bio-metal-organic framework. Results showed that bio-MOF-11 has higher permeability and selectivity than traditional zeolite for the separation of mixtures of these three gases. However, from the above studies, it is found that the diffusion of gas and pure liquid in MOF is mainly concentrated, and the reports on the diffusion of refrigerant in MOFs are still limited. Therefore, it is necessary to study the adsorption phenomenon and diffusion characteristics of mixtures in MOF.

Refrigerant is the energy carrier in ORC systems. There are various options of refrigerants for ORC and refrigeration system. Based on their environmental protection and excellent thermophysical properties, R32 and R1234yf have good prospects in refrigeration, ORC and other engineering fields [36–38]. Consequently, in this work, molecular dynamics simulations of the adsorption and diffusion of R32, R1234yf, and their mixtures in MOF-200 systems were performed. The results obtained may contribute to deeply understand the microscopic behavior of refrigerants in the MOFs and provide theoretical basis for experiments.

## 2. Materials and Methods

### 2.1. Molecular Model of Adsorption and Diffusion

The adsorption simulated system is shown in Figure 1, which consists of two parts, an organic working medium and metal–organic frameworks (MOF-200). The lattice parameters of MOF-200 are set to a = b = c = 51.79 Å, and α = β = γ = 90°. The mole ratios of the R32/R1234yf mixed refrigerant systems in this paper are shown in Table 1. The initial pressure is 2 MPa, the universal force field was chosen to depict the interatomic interactions in the simulations, and the Nose–Hoover method was employed to control the temperature of the system. The potential parameters for R32 and R1234yf are listed in Table 2.

**Table 1.** Number of molecules in mixed refrigerant system.

| Type | R32 | R32:R1234yf = 3:1 | R32:R1234yf = 1:1 | R32:R1234yf = 1:3 | R1234yf |
|------|-----|-------------------|-------------------|-------------------|---------|
| R32 | 2000 | 1500 | 1000 | 500 | 0 |
| R1234yf | 0 | 500 | 1000 | 1500 | 2000 |

**Table 2.** Force field parameters for R32, R1234yf.

| Atom | $\varepsilon$ (kcal/mol) | $\sigma$ (Å) | q (e) |
|------|--------------------------|--------------|-------|
| R32 | | | |
| C | 0.1085 | 3.15 | 0.43960 |
| H | 0.0157 | 2.2293 | 0.04158 |
| F | 0.0874 | 2.94 | −0.26138 |
| R1234yf | | | |
| C=(CH2) | 0.41000 | 3.4 | −0.4191 |
| C=(CF) | 0.41000 | 3.4 | 0.1974 |
| C-(CF3) | 0.31091 | 3.4 | 0.6306 |
| H | 0.06570 | 2.65 | 0.20473 |
| F(C=) | 0.23617 | 2.90 | −0.18254 |
| F(C-) | 0.23617 | 2.94 | −0.21196 |

| Type | Force | Force constant | |
|------|-------|----------------|---|
| R32 | Bond | $k_d$ (kJ mol$^{-1}$ Å$^{-2}$) | $r_0$ (Å) |
| | C-F | 1544.61 | 1.369 |
| | C-H | 1472.89 | 1.094 |
| | Angle | $k_\varphi$ (kJ mol$^{-1}$ rad$^{-2}$) | $\theta_0$ (deg) |
| | H-C-H | 146.54 | 113.6 |
| | F-C-F | 367.61 | 108.7 |
| | H-C-F | 249.92 | 108.6 |
| R1234yf | Bond | $k_d$ (kJ mol$^{-1}$ Å$^{-2}$) | $r_0$ (Å) |
| | C=C | 2831.69 | 1.331 |
| | C-C | 1328.84 | 1.511 |
| | C-H | 1627.07 | 1.086 |
| | -C-F | 1544.61 | 1.353 |
| | =C-F | 1864.73 | 1.330 |
| | Angle | $k_\varphi$ (kJ mol$^{-1}$ rad$^{-2}$) | $\theta_0$ (deg) |
| | H-C=C | 152.09 | 120.6 |
| | F-C-F | 367.61 | 107.5 |
| | F-C-C(=) | 313.17 | 111.3 |
| | H-C-H | 122.63 | 118.7 |
| | C=C-F | 211.38 | 122.6 |
| | F-C(=)-C | 319.57 | 112.5 |
| | C=C-C | 209.70 | 124.1 |
| | HC-C-HC | 146.54 | 113.6 |
| | HC-C-FC | 249.92 | 108.6 |

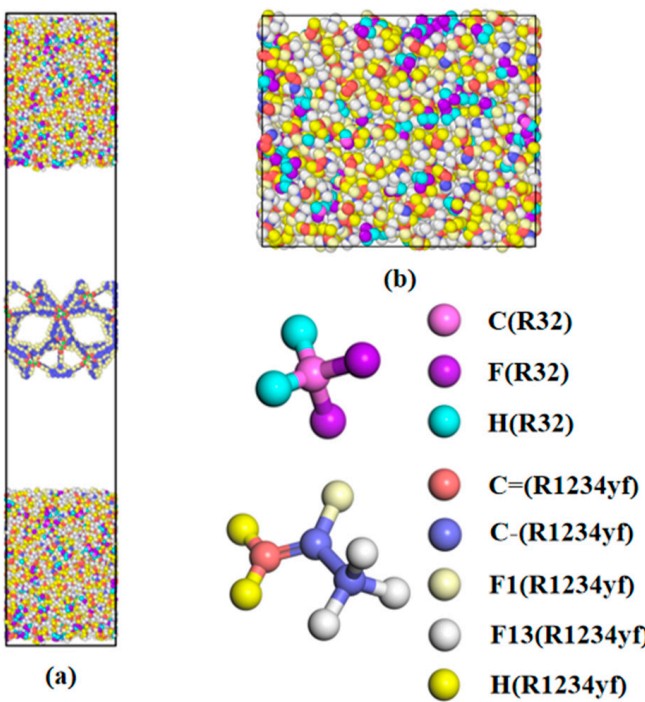

**Figure 1.** Initial composition of the simulation system. (**a**) Adsorption model; (**b**) diffusion model.

### 2.2. Simulation Details

In this study, the molecular simulation process includes adsorption and diffusion. The simulations were performed using LAMMPS (Largescale Atomic/Molecular Massively Parallel Simulator,2019,Albuquerque, New Mexico, USA) package. The cutoff distance and the timestep were set as 12 Å and 1 fs (10–15 s). The long-range Coulombic interaction was solved by PPPM method with the accuracy of 10-4. The equation of particles motion was integrated by the velocity Verlet algorithm, respectively.

The simulation process can be divided into three procedures: equilibrium, adsorption and diffusion. The equilibrium of the molecular model is a prerequisite for subsequent simulation. The entire system was simulated at different temperature in the NVT ensemble using the Nose–Hoover thermostat in the equilibrium stage. The simulation time is 1 ns ($10^{-9}$ s), which ensured that the liquid film was uniformly distributed in space at both ends of the simulation box. In the adsorption simulation, MOF-200 was fixed, and the boundary conditions were periodic in the X and Y directions. The adsorption stage lasted for 3 ns. Meanwhile, the number of adsorbed molecules and energy changes of MOF-200 were calculated. Next, to discuss the diffusion behavior of R32/R1234yf in MOF-200, we performed kinetic simulations at different loads and temperatures in the NVT ensemble. The optimized system is simulated with 2 ns. The three directions employ periodic boundary conditions.

## 3. Results

### 3.1. Adsorption Capacity

The adsorption performance of MOF structure can be characterized by adsorption capacity, adsorption heat, adsorption energy and other indicators. In order to more intuitively describe the adsorption process of the molecules in MOF-200, Figure 2 shows the number of molecules adsorbed in the model after the adsorption, which is converted into unit of g/g in this paper to represent the adsorption capacity. To demonstrate the superiority of MOF-200, we also compared the adsorption of R32/R1234yf in MOF-200 and ZN-MOF-74 in the literature, as shown in Table 3. As can be seen from the table, in pure fluid, the adsorption amount in MOF-200 is much greater than that of Zn-MOF-74 at 290 K.

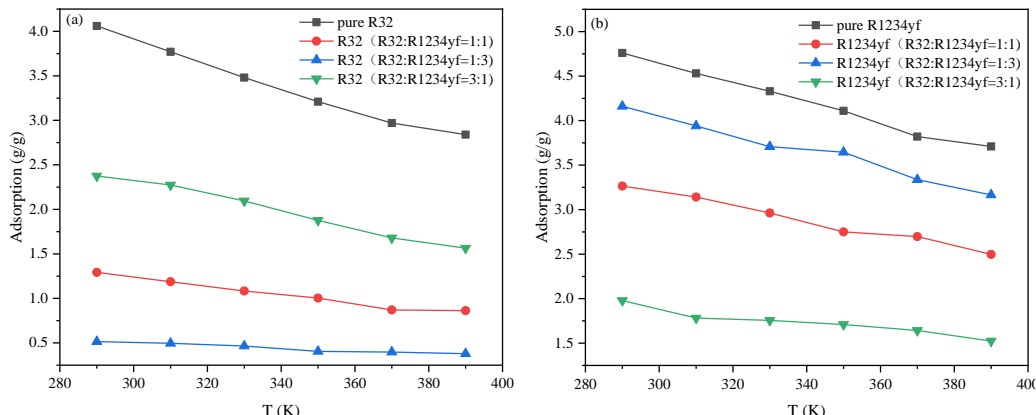

**Figure 2.** Adsorption capacity of R32 and R1234yf in MOF-200 at different temperatures. (**a**) R32; (**b**) R1234yf.

**Table 3.** Adsorption capacity of R32 and R1234yf in MOF-200 and Zn-MOF-74.

|  | **MOF-200** | **Zn-MOF-74** |
|---|---|---|
| R32 (g/g) | 4.06 | 0.624 |
| R1234yf (g/g) | 4.76 | 0.513 |

As shown in Figure 2, the number of adsorbed molecules of R32 in MOF-200 per unit mass is higher than that of R1234yf in the pure fluid adsorption system. This is mainly because the molecular structure of R32 is smaller and can effectively use the space in the pores of MOF-200 during the adsorption process. In the pure fluid adsorption system, we can also see that temperature is one of the most dominant influences on adsorption in the higher temperature range, as reflected by the fact that the gap between two adjacent adsorption lines becomes larger for every 20 K increase. Compared with R1234yf, the adsorption capacity of R32 varies greatly with temperature. In addition, the adsorption of R32/R1234yf mixtures occupies the pore space of MOF-200 together, resulting in less adsorption of each working medium than that of the pure working medium.

Adsorption selectivity is a criterion used to evaluate the adsorption performance of adsorbent in binary mixture. If the adsorption selectivity value is greater than 1, it indicates that the component is preferentially adsorbed in the binary mixture over the other components. The adsorption selectivity of R32 over R1234yf in MOF-200 is defined as follows:

$$s^a_{R32} = \frac{\left(x_{R32}/x_{R1234yf}\right)_{adsorbed}}{\left(x_{R32}/x_{R1234yf}\right)_{bulk}} \tag{1}$$

where $x_i$ denotes the average mole fraction of component i, the subscripts $(\ )_{adsorbed}$ and $(\ )_{bulk}$ mean the quantities of adsorbed molecules in pores and bulk phases, respectively. It is calculated the influence of three mole ratios on R32/R1234yf adsorption selectivity at different temperatures, as shown in Figure 3.

Figure 3 shows that when the mole ratio of R32/R1234yf is 3:1, the adsorption selectivity coefficients of R32 over R1234yf are all greater than 1. That means that the adsorption capacity of R32 is better than that of R1234yf in MOF-200, which is consistent with the above results. When the molar ratio is equal, the selectivity coefficient of R32 over R1234yf adsorption is less than 1, indicating that R1234yf is more conducive to adsorption in the mixture, which is different from the adsorption of the pure working medium. The selectivity coefficients of the three mole ratios all increase first and then decrease, implying that the maximum selectivity exists in the temperature range from 310 K to 330 K.

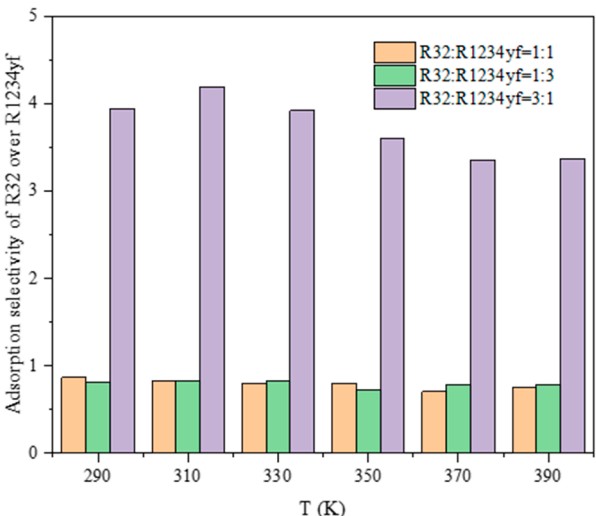

**Figure 3.** Adsorption selectivity of R32 over R1234yf at different systems.

### 3.2. Adsorption Heat

Adsorption heat is an important indicator in the study of the adsorption process, which mainly reflects the strength of the force between adsorbent and adsorbate. Theoretically, the isostatic adsorption heat, $q_{st}$ is approximated by

$$\left(\frac{\ln p}{\delta T}\right)_\theta = \frac{q_{st}}{RT^2} \tag{2}$$

where $\theta$ is the fluid coverage on the solid surface, $P$ is the adsorption saturation pressure, $T$ is the adsorption temperature, $R$ is the gas constant and $q_{st}$ is the equivalent heat of adsorption, respectively.

The value of adsorption heat is shown in Table 4. At different temperatures, the isostatic adsorption heat of R32/R1234yf in MOF-200 is less than 42 kJ/mol [34], so it belongs to physical adsorption. For both pure and mixed refrigerants, the adsorption heat of R32 in MOF-200 is smaller than that of R1234yf. This is because the molecular size and molecular weight of R32 are smaller than that of R1234yf, which requires less heat to achieve saturation adsorption. The isostatic adsorption heat is also used to describe the strength of the interaction between adsorbents and adsorbates. As can be seen from the values in the table, the interaction between R32 and MOF-200 is weakened after the mixing of the two refrigerants, while the influence of R1234yf is small.

**Table 4.** The isostatic adsorption heat in MOF-200.

| Type | $q_{st}$ (kJ/mol) |
|---|---|
| R32 | 24.5 |
| R1234yf | 25.8 |
| R32 (R32:R1234yf = 1:1) | 17.32 |
| R1234yf (R32:R1234yf = 1:1) | 21.74 |
| R32 (R32:R1234yf = 1:3) | 11.78 |
| R1234yf (R32:R1234yf = 1:3) | 24.39 |
| R32 (R32:R1234yf = 3:1) | 19.45 |
| R1234yf (R32:R1234yf = 3:1) | 20.62 |

### 3.3. Self-Diffusivity Coefficients of R32-R1234yf in MOF-200

Diffusion is a thermally activated process in which migrating atoms cross an energy potential barrier and move from a local minimum energy position to a neighboring null position. The self-diffusion coefficient directly related to the molecular motion can be

obtained from Einstein's diffusion law. To gain insight into the diffusion capacity of the organic working medium in MOF-200, the self-diffusion coefficients of R32/R1234yf at different temperatures are illustrated in Figure 4.

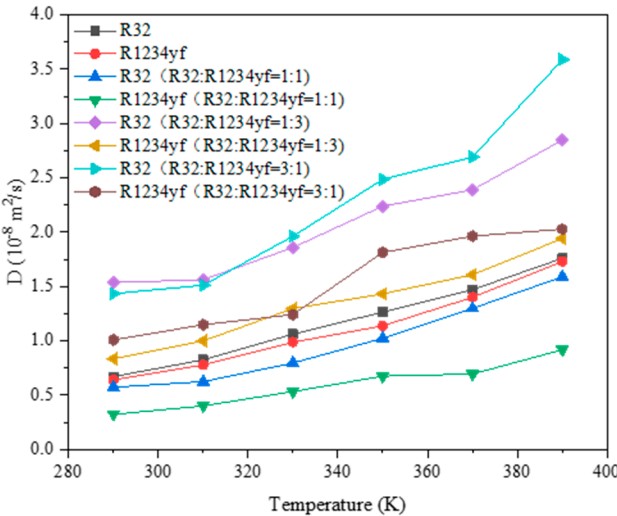

**Figure 4.** The self-diffusion coefficients of R32/R1234yf at different temperatures.

It can be seen that the self-diffusion coefficient of R32 in MOF-200 is larger than that of R1234yf in MOF-200 because the molecular structure of R32 is smaller, then the diffusion will be faster. As the temperature increases, the self-diffusion coefficient also increases. This is because the higher the temperature, the more the collision of the molecules will increase. According to the principle of conservation of energy, the internal energy of the molecule can be converted into kinetic energy, increasing the molecule's motion and making it easier to diffuse. Previous studies have proven that the diffusion coefficient of fluid molecules is negatively relevant to adsorption heat [39], that is, the greater the adsorption heat of fluid molecules, the stronger the interaction force between them and the pore wall, and the smaller the diffusion coefficient. Compared with the pure fluid, the self-diffusion coefficients of R1234yf and R32 are smaller when the molar ratio of R32/R1234yf is 1:1.

Meanwhile, the diffusion selectivity is a criterion used to evaluate the preferential diffusion ability of one species over others in porous media, as defined below:

$$s_{R32}^d = \frac{D_{R32}^{self}}{D_{R1234yf}^{self}} \tag{3}$$

where $D_{R32}^{self}, D_{R1234yf}^{self}$ are the self-diffusion coefficients (m$^2$/s) of R32 and R1234yf.

The diffusivity selectivity of R32 over R1234yf in MOF-200 is shown in Figure 5. All values of diffusivity selectivity in the Figure 5 are greater than 1, indicating that R32 diffused in MOF-200 preferentially to R1234.

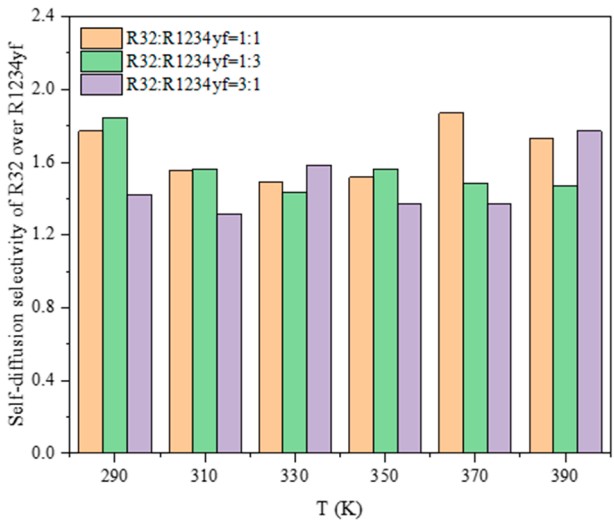

**Figure 5.** Diffusivity selectivity of R32 over R1234yf in MOF-200.

The diffusion of R32/R1234yf in MOF-200 is an activation process. The diffusion activation energy of R32/R1234yf in MOF-200 can be calculated by the Arrhenius equation.

$$D = D_0 \exp\left(-\frac{E_a}{RT}\right) \tag{4}$$

where $D_0$ is the pre-exponential factor, $E_a$ is the activation energy, and $R$ is the gas molar constant. The results are presented in Figure 6 for R32 and R1234yf in MOF-200, respectively.

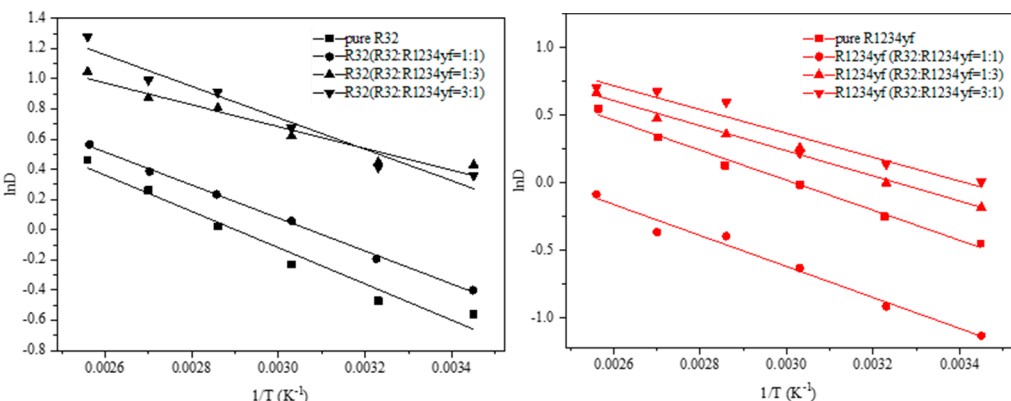

**Figure 6.** Fitting curves of activation energy of lnD and $T^{-1}$.

According to Equation (4), the self-diffusion activation energy Ea of R32 and R1234yf in MOF-200 was obtained as shown in Table 5. From the table, it can be noticed that the diffusion activation energies of R32 and R1234yf are the largest when the molar ratio is 1:1. This indicates that there is a higher energy barrier in the pore channel, resulting in the smallest diffusion coefficients in the system, which is consistent with the results described previously.

**Table 5.** Fitting results of R32/R1234yf diffusion activation energy.

| Types | Ea (kJ/mol) | $D_0$ (m$^2$/s) |
|---|---|---|
| R32 | 9.078 | 28.587 |
| R1234yf | 9.220 | 28.383 |
| R32 (1:1) | 9.128 | 33.316 |
| R1234yf (1:1) | 9.590 | 17.133 |
| R32 (1:3) | 6.046 | 17.530 |
| R1234yf (1:3) | 7.759 | 20.806 |
| R32 (3:1) | 7.728 | 48.903 |
| R1234yf (3:1) | 8.381 | 20.663 |

## 4. Conclusions

In summary, the adsorption and diffusion properties of different molar ratios of R32/R1234yf in MOF-200 were studied by molecular simulation. The simulated conclusions are summarized as follows. In the pure fluid adsorption system, the number of adsorbed molecules of R32 in MOF-200 per unit mass is higher than that of R1234yf. It can also be seen that temperature has a greater effect on adsorption in the higher temperature range. Due to competitive adsorption, the adsorption capacity of the mixture in MOF-200 is lower than that of the pure fluid. The diffusion activation energy of R32 is less than that of R1234yf, which explains why the self-diffusion coefficient of R32 in MOF-200 is larger than that of R1234yf. The diffusion activation energies of R32 and R1234yf are the largest when the molar ratio is 1:1. This indicates that there is a higher energy barrier in the pore channel, resulting in the smallest diffusion coefficients in the system.

Furthermore, one of the remaining challenges in studying molecular diffusion in MOFs materials is to understand in detail the difference between self-diffusion and transport diffusion at the microscopic level. In order to more accurately and reliably analyze the diffusion characteristics of organic working medium in MOFs, the transfer diffusion coefficient is worth studying further.

**Author Contributions:** Conceptualization; methodology, software, validation, formal analysis, investigation, resources, data curation; writing—original draft preparation, B.J.; writing—review and editing, D.X., G.W. All authors have read and agreed to the published version of the manuscript.

**Funding:** This research was funded by the National Natural Science Foundation of China, grant number 51506013.

**Institutional Review Board Statement:** Not applicable.

**Informed Consent Statement:** Not applicable.

**Data Availability Statement:** Not applicable.

**Acknowledgments:** We acknowledge the support of the National Natural Science Foundation of China.

**Conflicts of Interest:** The authors declare no conflict of interest.

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
