# Peer review of "Adsorption and Self-Diffusion of R32/R1234yf in MOF-200 Nanoparticles by Molecular Dynamics Simulation"

_processes, doi:10.3390/pr10091714_

Round 1

Reviewer 1 Report

Title: “Adsorption and Self-Diffusion of R32/R1234yf in MOF-200 nanoparticles by molecular dynamics simulation”

After reviewing the present manuscript, reviewer found that the authors made interesting work and required analysis for the “Adsorption and Self-Diffusion of R32/R1234yf in MOF-200 nanoparticles by molecular dynamics simulation”. The reviewer found that this manuscript is fit with Processes MDPI and has following general comments and specific comments with regards to the improvement of the manuscript prior to the publication.

The manuscript is written in good English despite some minor errors. The content of the manuscript is well organized, and length and depth of the work is fair and sufficient for an article. This manuscript can be recommended for a publication after attending to suggestions below.

Be consistent with the space between units and numbers. Use of stops in appropriate places and correct way of referencing must be carefully looked.

The comments below are organized in order as they appear in the manuscript.

1.           Graphical abstract not included- Adding a graphical abstract would help add weightage to the manuscript.

2.           Include highlights to the manuscript, to present the most significant findings of the research.

3.           Keywords-Each keyword should start with an upper case.

4.           Page2-Para1- How can authors justify this statement? Can you add results including figure and a table for large pore size (SEM), adsorption, and desorption isotherm? “According to previous works, MOF-200 has good adsorption effect on fluid due to its large pore size, high BET and Langmuir specific surface area”

5.           There are published papers on Adsorption and Self-Diffusion of R32/R1234yf in MOF: The authors need to summarize those studies in the introduction section and thus point out the knowledge gaps and why this work is necessary.

6.           Is that possible for authors to add a comparison table with different adsorbents and present work.

Author Response

Reviewer #1:

After reviewing the present manuscript, reviewer found that the authors made interesting work and required analysis for the“Adsorption and Self-Diffusion of R32/R1234yf in MOF-200 nanoparticles by molecular dynamics simulation”. The reviewer found that this manuscript is fit with Processes MDPI and has following general comments and specific comments with regards to the improvement of the manuscript prior to the publication.

The manuscript is written in good English despite some minor errors. The content of the manuscript is well organized, and length and depth of the work is fair and sufficient for an article. This manuscript can be recommended for a publication after attending to suggestions below.

Be consistent with the space between units and numbers. Use of stops in appropriate places and correct way of referencing must be carefully looked.

Reply: Thanks for reviewing our paper and providing useful comments.

The comments below are organized in order as they appear in the manuscript.

* Graphical abstract not included- Adding a graphical abstract would help add weightage to the manuscript.

Reply:

Thanks for your suggestion.

We have added a graphic summary in the abstract, which contains the adsorption diffusion system and self-diffusion coefficient diagram of this paper.

* Include highlights to the manuscript, to present the most significant findings of the research.

Reply:

Thanks for the suggestion.

Understanding the adsorption-diffusion process of the mixture in MOFs at the molecular level is important to further improve the efficiency of the Organic Rankine cycle. In this study, the adsorption phenomenon and diffusion characteristics of refrigerant R32/R1234yf in MOF-200 were studied. The microscopic behavior of the molecules in the MOF structure was thoroughly discussed, and the diffusion performance of the refrigerant mixtures in the MOFs was analyzed for the first time.

* Keywords-Each keyword should start with an upper case.

Reply:

Thanks for the reviewer’s comments.

We modified the keywords as “Refrigerants; Adsorption; Diffusion; Nanofluids; Molecular dynamics”.

* Page2-Para1- How can authors justify this statement? Can you add results including figure and a table for large pore size (SEM), adsorption, and desorption isotherm? “According to previous works, MOF-200 has good adsorption effect on fluid due to its large pore size, high BET and Langmuir specific surface area”

Reply:

Thanks for the reviewer’s comments.

According the extensive literature review, we mentioned that García simulated the adsorption performance of refrigerants in 40 experimentally the available MOFs in this paper. Among them, the simulation result of MOF-200 was better, which he speculated was due to the large pore size and high BET of MOF-200.

Xu also studied the adsorption performance of CO2 in MOF-200.It lists the structural information of MOF-200, as shown in the following table.It can be seen from the table that MOF-200 has good adsorption effect on fluid due to its large pore size, high BET and Langmuir specific surface area.

Material

Unit cell(Å)

Cell angle(°)

Density(g/cm3)

Surface(Å23)

a

b

c

α

β

γ

MOF-200

52.02

52.02

42.32

90

90

120

0.22

0.10

* There are published papers on Adsorption and Self-Diffusion of R32/R1234yf in MOF: The authors need to summarize those studies in the introduction section and thus point out the knowledge gaps and why this work is necessary.

Reply:

Thanks for the reviewer’s comments. We followed the suggestion.

We added “ Previous studies mainly focused on the discussion of the adsorption phenomenon of pure fluids in MOF, and there were few reports on the study of mixtures in MOF.” 

We also added “However,from the above studies, it is found that the diffusion of gas and pure liquid in MOF is mainly concentrated, and the reports on the diffusion of refrigerant in MOFs are still limited.Therefore, it is necessary to study the adsorption phenomenon and diffusion characteristics of mixtures in MOF.”in the Introduction.

* Is that possible for authors to add a comparison table with different adsorbents and present work.

Reply:

Thanks for the reviewer’s comments.

We added a comparison table with different adsorbents and present work. Since the central metal of MOF-200 is zinc, Zn-MOF-74 was selected from the literature for comparison. As can be seen from the table, in pure fluid, the adsorption amount in MOF-200 is much greater than that of Zn-MOF-74 at 290K.

Table 3 Adsorption capacity of R32 and R1234yf in MOF-200 and Zn-MOF-74

MOF-200

Zn-MOF-74

R32(g/g)

4.06

0.624

R1234yf(g/g)

4.76

0.513

Thanks again for your comments.

Reviewer 2 Report

The authors have used the molecular dynamics simulation to calculate the adsorption and diffusion properties of different molecules with different molar ratios in order to understand the properties of MOFs. The authors must address these comments.

1. The authors have discussed so much on the molecular size and weight of the molecular systems they have considered for the simulation study. But no where there is any mention of the composition of the two systems, R32 and R1234yf. It is very important to give details for the same including how the molecules are bonded in the respective systems.

2. In eq. 2, why did the authors consider 'small del' for the temperature 'T' instead of 'exact differential delta'? I would have expected the latter one since temperature is a state function. Another point, the authors are writing 'Qst' and 'qst' both to denote equivalent heat of adsorption. Please maintain consistency.

3. In the line 'As the temperature increases, the self-diffusion coefficient also increases. This is because the higher the temperature, the collision and diffusion of the molecules will increase.' the authors say that the self diffusion increases because diffusion increases. It is more or less the same thing, instead, the authors are encouraged to explain as to why do the self-diffusion increases? Will the molecular weight have any contribution towards it? How the migration energy may differ for those two cases?

4. The authors have just mentioned the method of calculating the diffusion parameters as 'Einstein's diffusion law'. But, there is no explanation of this equation and also, no method describing the calculation.

5. There should be a space between the numeric figure and the unit. Also, the units have not been mentioned properly, for eg. Kcal should be written as kcal. This issue is repetitive at several places throughout the text. 

6. The authors should clearly define the importance of 'yf'.

7. What can the authors deduce about the thermophysical properties of MOFs-modified refrigerants from the theoretically estimated adsorption and diffusion properties of molecular systems?

Author Response

Reviewer #2:

The authors have used the molecular dynamics simulation to calculate the adsorption and diffusion properties of different molecules with different molar ratios in order to understand the properties of MOFs. The authors must address these comments.

Reply: Thanks for reviewing our paper and providing useful comments.

*The authors have discussed so much on the molecular size and weight of the molecular systems they have considered for the simulation study. But no where there is any mention of the composition of the two systems, R32 and R1234yf. It is very important to give details for the same including how the molecules are bonded in the respective systems.

Reply:

Thanks for the reviewer’s comments.

We have mentioned the composition of the two systems, R32 and R1234yf in the Molecular Model of Adsorption and Diffusion. We gave the molecular structure of R32/R1234yf and the composition of the system in Figure 1, and the molar ratio of R32/R1234 in the mixed refrigerant system is also given in Table 2.

Fig. 1. Initial composition of the simulation system. (a) adsorption model; (b) diffusion model

Table 1 Number of molecules in mixed refrigerant system

Type

R32

R32:R1234yf=3:1

R32:R1234yf=1:1

R32:R1234yf=1:3

R1234yf

R32

2000

1500

1000

500

0

R1234yf

0

500

1000

1500

2000

*In eq. 2, why did the authors consider 'small del' for the temperature 'T' instead of 'exact differential delta'? I would have expected the latter one since temperature is a state function. Another point, the authors are writing 'Qst' and 'qst' both to denote equivalent heat of adsorption. Please maintain consistency.

Reply:

Thanks for the reviewer’s comments.

In this paper, the heat of adsorption is solved assuming that the fluid molecules reach saturation adsorption in the MOFs, so the isostatic adsorption heat can be solved by the Clausius-Clapeyron equation. The equation is

We revised the manuscript to maintain the consistency.

* In the line 'As the temperature increases, the self-diffusion coefficient also increases. This is because the higher the temperature, the collision and diffusion of the molecules will increase.' the authors say that the self diffusion increases because diffusion increases. It is more or less the same thing, instead, the authors are encouraged to explain as to why do the self-diffusion increases? Will the molecular weight have any contribution towards it? How the migration energy may differ for those two cases?

Reply:

Thanks for the reviewer’s comments.

When temperature is a variable, the increase of self-diffusion is due to the increase of molecular kinetic energy. At high temperatures, heat is absorbed by molecules and converted into kinetic energy, making them more active,thus the self-diffusion increases.

The mass of molecules also has a certain effect on the diffusion. When the molecules are larger, their velocity will decrease. Both of these factors have an effect on the speed of the molecules, thus changing the migration energy of molecular diffusion

* The authors have just mentioned the method of calculating the diffusion parameters as 'Einstein's diffusion law'. But, there is no explanation of this equation and also, no method describing the calculation.

Reply:

Thanks for the reviewer’s comments. We followed the suggestion.

We added the reference about “Einstein's diffusion law”. The equation is

where T is the temperature (K), N is the number of molecules, t is the simulation time, ri (0) is the initial coordinate position of the i molecule, and ri (t) expresses the coordinate position of the i molecule at time t.

* There should be a space between the numeric figure and the unit. Also, the units have not been mentioned properly, for eg. Kcal should be written as kcal. This issue is repetitive at several places throughout the text.

Reply:

Thanks for the reviewer’s comments. We followed the suggestion.

We revised the units in the figure.

* The authors should clearly define the importance of 'yf'.

Reply:

Thanks for the reviewer’s comments.

R1234yf is a hydrofluorocarbon refrigerant,which has good prospects in refrigeration, ORC and other engineering fields.

*What can the authors deduce about the thermophysical properties of MOFs-modified refrigerants from the theoretically estimated adsorption and diffusion properties of molecular systems?

Reply:

Thanks for the reviewer’s comments. We followed the suggestion.

The simulation method used in this paper has been verified in previous studies. According to previous studies, it can be concluded that the combination of refrigerant and MOF enhances the heat transfer properties of the working fluid.Reports of anomalous increases in specific heat and heat transfer coefficients appeared early in the nanofluid literature. What's more,Edder et al. also reported systematic studies on the adsorption of refrigerant in MOF, showing that adsorption-based refrigeration cycles facilitate renewable energy applications and allow for energy storage.

Thanks again for your comments.

Round 2

Reviewer 2 Report

The authors have responded to all the comments satisfactorily.